# The Association between Symptomatic Rotavirus Infection and Histo-Blood Group Antigens in Young Children with Diarrhea in Pretoria, South Africa

**DOI:** 10.3390/v14122735

**Published:** 2022-12-08

**Authors:** Kebareng Rakau, Maemu Gededzha, Ina Peenze, Pengwei Huang, Ming Tan, Andrew Duncan Steele, Luyanda Mapaseka Seheri

**Affiliations:** 1Diarrheal Pathogens Research Unit, Department of Virology, Sefako Makgatho Health Sciences University, Medunsa 0204, South Africa; 2Department of Immunology, Faculty of Health Sciences, University of Witwatersrand and National Health Laboratory Service, Johannesburg 2001, South Africa; 3Division of Infectious Diseases, Cincinnati Children’s Hospital and Medical Center, Cincinnati, OH 45245, USA; 4Department of Pediatrics, College of Medicine, University of Cincinnati, Cincinnati, OH 45245, USA

**Keywords:** rotavirus, HBGA, Lewis antigens, South Africa

## Abstract

Objectives: Recently, histo-blood group antigens (HBGAs) have been identified as receptors or attachment factors of several viral pathogens. Among rotaviruses, HBGAs interact with the outer viral protein, VP4, which has been identified as a potential susceptibility factor, although the findings are inconsistent throughout populations due to HBGA polymorphisms. We investigated the association between HBGA phenotypes and rotavirus infection in children with acute gastroenteritis in northern Pretoria, South Africa. Methods: Paired diarrheal stool and saliva samples were collected from children aged ≤ 59 months (n = 342) with acute moderate to severe diarrhea, attending two health care facilities. Rotaviruses in the stool samples were detected by commercial EIA and the rotavirus strains were characterized by RT-PCR targeting the outer capsid VP7 (G-type) and VP4 (P-type) antigens for genotyping. Saliva-based ELISAs were performed to determine A, B, H, and Lewis antigens for blood group typing. Results: Blood type O was the most common blood group (62.5%) in this population, followed by groups A (26.0%), B (9.3%), and AB (2.2%). The H1-based secretors were common (82.7%) compared to the non-secretors (17.3%), and the Lewis antigen positive phenotypes (Le^(a+b+)^) were predominant (54.5%). Blood type A children were more likely to be infected by rotavirus (38.8%) than any other blood types. P[4] rotaviruses (21/49; 42.9%) infected only secretor individuals, whereas P[6] rotaviruses (3/49; 6.1%) only infected Le^(a−b−)^, although the numbers were very low. On the contrary, P[8] rotaviruses infected children with a wide range of blood group phenotypes, including Le^(a−b−)^ and non-secretors. Conclusions: Our findings demonstrated that Lewis antigens, or the lack thereof, may serve as susceptibility factors to rotaviral infection by specific VP4 genotypes as observed elsewhere. Potentially, the P[8] strains remain the predominant human VP4 genotype due to their ability to bind to a variety of HBGA phenotypes.

## 1. Introduction

Globally, rotavirus is the leading cause of severe gastroenteritis in children under 5 years of age. Although two vaccines, Rotarix^®^ (GSK Biologicals, Rixensart, Belgium) and RotaTeq^®^ (Merck & Co, White River, PA, USA) have been recommended for use since 2009 by the World Health Organization (WHO) and are utilized in over 100 countries, rotavirus is still responsible for an estimated 128,500 deaths globally every year [1]. The majority of rotavirus deaths are in low-income countries in Asia and sub-Saharan Africa [1]. Since the initial WHO recommendation of the two vaccines, two new rotavirus vaccines developed in India have been WHO pre-qualified in 2018 [2] and are now being introduced in African countries with Gavi support. 

Rotavirus is a dsRNA virus with a genome comprised of eleven segments coding for six structural and six non-structural proteins [3]. Two of the structural proteins, viral protein (VP) 7 and VP4, form the outer capsid of the virus and are involved in the attachment and entry of rotavirus into the host cell, and are recognized as neutralization antigens. Structurally, VP4 forms the spikes that protrude from the VP7 layer of rotavirus, and can be cleaved by trypsin into two proteins, VP5* and VP8* [4]. The VP8* that interacts with glycan receptors is responsible for the attachment of rotavirus to the host cells, while VP5* is involved in the penetration of the virus inside the membrane of a host cell [4,5,6]. Based on the sequences of the genes encoding VP7 and VP4, rotaviruses are classified into different G and P genotypes, respectively. Of these, VP7 G1-G4, G9, and G12, in combination with VP4 P[4], P[6], and P[8], cause the majority of human rotavirus diarrhea cases globally [7], including in Africa.

Rotavirus infects the epithelial cell lining of the gastrointestinal tract via recognition of specific carbohydrates as receptors or attachment factors [8]. The attachment of rotavirus to the host epithelial lining is complex. Recent studies have implicated histo-blood group antigens (HBGAs), a group of fucose-containing carbohydrates, to function as receptors for rotavirus attachment [9]. HBGAs distribute abundantly on the mucosal surface of the intestinal tract as well as in biological fluids such as saliva and breast milk [10,11]. The syntheses of H and Lewis antigens are regulated by fucosyltransferase (FUT) 2 and FUT 3, while production of A or B antigens are regulated by enzymes A or B, resulting in various HBGA phenotypes, including ABO blood types, H type 1, and Lewis antigens, respectively [12,13]. 

The *FUT2* gene encodes the α1,2 fucosyltransferase (FUT2) enzyme that catalyzes the production of the H type 1 antigen by adding fucose to the α1,2 linkage of the type I precursor. On the other hand, the *FUT3* gene encodes the α1,3/4 fucosyltransferase (FUT3) enzyme that facilitates the production of Lewis antigens, including Lewis a (Le^(a+)^) and Lewis b (Le^(b+)^), by adding fucose to the α1,4 linkage of the type I precursor and H type 1 antigen, respectively. In approximately 80% of the global population, the *FUT2* gene is functional and the FUT2 enzyme is active, leading to production of H-related antigens, including H type 1, and/or Le^(b+)^, and such HBGA phenotypes are called secretor types [10,13]. By contrast, in some cases, mutations occur in *FUT2* genes, leading to inactivation of the FUT2 enzyme. As result, the H type 1 related antigens cannot be produced, leaving only Le^(a)^ positive phenotypes, which are defined as non-secretor types. Different phenotypes of the Lewis antigen profiles exist, including Le^(a+b−)^ (non-secretors), Le^(a−b+)^ (secretors) and Le^(a+b+)^ (secretors), or Le^(a−b−)^, which could either be secretor or non-secretor [14].

The majority of epidemiological studies investigating the association of rotavirus infection or vaccination outcome with HBGAs have reported that P[8] and P[4] rotaviruses have a predilection to infect HBGA secretors, and that non-secretors may be more resistant to P[8] rotavirus infection [9,15,16,17,18,19,20]. Only a single study conducted in Tunisia observed different results in which both non-secretors and secretors were readily infected with P[8] rotaviruses [21]. Rotavirus P[6] strains are reported to occur at a higher prevalence in non-secretors [17], which has led to speculation that the lower efficacy observed in African populations with the VP4 P[8]-based vaccines, such as Rotarix and RotaTeq, may be associated with a higher distribution of non-secretors in the population [22]. It has also been postulated that the higher circulation of P[6] strains in Africa may be linked to the high rates of Lewis-negative individuals in the population [17,23]. Interestingly, recent studies have identified a structural mechanism by which the rotavirus VP4 P[8] interacts with specific binding domains on the H1 antigen, giving an explanation of this important interaction [24]. 

We therefore decided to examine the association between HBGA profiles of children with rotavirus gastroenteritis (RVGE) in a cross-sectional study in northern Pretoria, South Africa.

## 2. Materials and Methods

### 2.1. Sample Population

Diarrheal surveillance in infants and young children < 5 years of age, has been ongoing in South Africa for over 30 years. Initially established to monitor rotavirus disease burden, and post-rotavirus vaccine introduction in 2009, to monitor the impact of the vaccine programme. More recently, this national surveillance program has focused on both rotavirus and other etiologies associated with acute watery diarrhea in young children attending health care facilities for treatment. 

The children enrolled in this study constituted part of this on-going diarrhea surveillance in the Diarrheal Pathogens Research Unit (DPRU) based at Sefako Makgatho Health Sciences University and Dr. George Mukhari Academic Hospital complex. Dr. George Mukhari Academic Hospital (DGMAH) is a tertiary-care 1700 bed hospital based 30 kms northwest of Pretoria in South Africa. It has 23 associated health clinics and District Level Hospitals. Oukasie Primary Health Care Clinic (OPHC) is an outpatient health-based clinic 23.6 kilometers from DGMAH and refers patients requiring admission to DGMAH.

This study utilized the ongoing diarrhea surveillance to investigate the association of HBGAs and rotavirus infection. Parents of children with diarrhea were offered participation in this study and gave informed consent. Between June 2015 and November 2017, paired stool and saliva samples were collected in a total of 342 children less than 5 years of age who were seeking medical assistance for acute diarrhea at either OPHC or DGMAH. Acute diarrheal disease as reported by the parents was a presenting symptom for inclusion in the study. Diarrhea was defined as three or more episodes within 24 h. 

### 2.2. Ethical Consideration

Ethical approval to conduct the study was obtained from Sefako Makgatho Health Sciences University Research Ethics Committee (SMUREC/P219/2015) and the management of OPHC and DGMAH. Paired diarrheal stools and saliva samples were collected from the children (under 59 months of age) whose parents/guardians gave consent. All the study methods were carried out guided by appropriate ethical guidelines and regulations.

### 2.3. Sample Collection

Diarrheal stool samples were collected from children on the day of enrolment. The stool samples were collected into 5 mL stool collection containers (B & M Scientific, Cape Town, South Africa) and stored immediately at 2–8 °C in refrigerators on site for later transport to the laboratory for testing. Stool samples at OPHC were transferred to the laboratory weekly, whereas samples at the Pediatric Department of DGMAH were transferred daily to the laboratory. EIAs for detection of rotavirus antigen are conducted monthly in the laboratory at the Diarrheal Pathogens Research Unit (DPRU) based at Sefako Makgatho Health Sciences University.

For saliva collection, a plain sterile oral swab (AEC-Amersham soc Ltd., Johannesburg, South Africa) was used to rub the inside of the child’s mouth on both cheeks and was placed under the tongue for 10 s to absorb saliva. Immediately thereafter, the swab was placed into a 4 mL cryovial tube (Carl Roth GmbH & co, Karlsruhe, Germany) containing 1 mL of distilled water and transported to the DPRU on ice where they were kept at −80 °C until testing.

### 2.4. Rotavirus Detection and Strain Characterization in Stool

The detection of rotavirus in stool was performed following the WHO manual of rotavirus detection and characterization [25]. Briefly, group A rotavirus antigens in stool samples were tested using the commercially available ProSpecT^TM^ rotavirus enzyme immunoassay (EIA) kit (Oxoid, Basingstoke, UK). Rotavirus dsRNA was extracted using Qiagen^®^ Viral RNA extraction kit (Qiagen, Hilden, Germany). Gene segment 4 that codes for VP4 and gene segment 9 which codes for VP7 were reverse transcribed using Avian Myeloblastosis Virus (AMV) reverse transcriptase. Two well-recognized sets of primers, *Con2/Con3* and *sBeg/End9*,, specific to VP4 (VP8* region) and VP7 were used to amplify the genes, respectively [25,26,27,28]. These amplicons were utilized to characterize rotavirus strains into G and P types. The following rotavirus genotypes were targeted: G1-G4, G8, G9, G10, G12 including P[4], P[6], P[8] and P[14], representing the most common human G and P types circulating in Sub-Saharan Africa [28]. 

### 2.5. HBGA Phenotyping in Saliva

The HBGA phenotype profiles of individuals were determined by saliva-based enzyme linked immuno-assays (ELISAs) using the corresponding monoclonal antibodies against individual HBGA glycans (A, B, H1, Le^a^, and Le^b^) as described previously [9]. Saliva samples were briefly boiled to inactivate proteinaceous factors and diluted 1000-fold. The diluted saliva specimens were then coated on the microtiter plates (Dynex Immulon; Dynatech, Franklin, MA) overnight. After blocking with non-fat milk, individual monoclonal antibodies specific to blood group antigens A (BG-2/T36, Covance Inc); B (BG-3/HE29, Accurate Chemical & Scientific Corp., Carle Place, USA); H1 (BG-4/17-206, Covance Inc); Le^a^, (BG-5/T174, Covance Inc); and Le^b^ (BG-6/T218, Covance Inc) at 1:100 dilution were added to the plates for 60 min incubation. After washing, corresponding secondary antibody-horseradish peroxidase (HRP) conjugates at dilutions (HRP-goat anti mouse IgG3 for A and H1, 1:2500; HRP-goat anti mouse IgG1 for Le^a^, 1:4000; HRP-goat anti mouse IgM for B, Le^b^, 1:2500) were added. All reagents were from Immunology Consultants Laboratory Inc., Newberg). 

The reactions were observed by adding HRP substrate reagents (optEIA, BD Bioscience, San Diego, CA, USA) and were read by spectra Max 340pc (Molecular Devices, San Jose, CA, USA). Four well-characterized saliva samples from our lab stock containing all interested HBGA types were included in each plate as internal controls. The cut-off of a positive signal was OD_450_ = 0.2. Based on HBGA biosynthesis pathway, different HBGA categories are defined as follows: (i) secretors are defined as those who have one or more H1 antigen-related HBGA types, including A, B, H1, and/or Le^b^ antigen; (ii) non-secretors are defined as those who lack the H1 antigen-related HBGA types and are positive with Le^a^ antigen; (iii) Le^(a−b+)^ individuals are Le^a^ negative but Le^b^ positive; (iv) Le^(a+b+)^ are those with both Le^a^ and Le^b^ antigens; (v) Le^(a+b−)^ are those with Le^a^ positive, but Le^b^ negative, and (vi) Le^(a−b−)^ individuals are those lacking Le^a^ and Le^b^ antigens.

### 2.6. Statistical Analysis

Data entry was carried out using MS Excel software (2019, Seattle, WA, USA). All statistical analysis were conducted using R software version 4.2.1 (RStudio, 2022, Boston, MA, USA). Descriptive statistics were used to summarize baseline characteristics. The absence or presence of rotavirus infection was considered the outcome of interest. Cross tabulations of characteristics and predictor variables with the dependent variable were constructed. Contingency tables were evaluated by Pearson’s chi-squared test for qualitative variables above 5 and Fisher’s exact test for qualitative variables below 5. 

After descriptive analysis, simple and multiple logistic regression models were explored. Predictor variables with a *p*-value less than 0.20 in the simple logistic regression models were initially considered for multivariable logistic regression model building based on Hosmer–Lemeshow test. To measure the strength of association, we obtained the crude odds ratio (cOR), using simple logistic regression models, and adjusted odds ratio (aOR), with 95% CI using multiple logistic regression models, for controlling possible confounding variables. The Hosmer_Lemeshow was used to test the goodness of fit for the multiple logistic regression models at a significance level of 0.05.

## 3. Results

### 3.1. Demographics 

The 342 recruited children were predominantly males (51.5%, 176/342), aged between 0 and 59 months (mean age of 13.2 months, standard deviation of 10.1 months), and belonged to the Black ethnic group (Table 1). Of these children, 101 were admitted overnight to DGMAH pediatric wards and 241 attended OPHC as outpatients. Clinically, these children exhibited acute diarrhea, accompanied with fever, vomiting, and refusal to eat. Based on the information retrieved from their Road to Health Card (RTHC), which provides information on the child’s health parameters and immunization records at the time of study recruitment, only 1.2% (4/342) of the children were unvaccinated with Rotarix vaccine; 6.7% (23/342) had received only one dose; and 82.7% (283/342) had received the full two doses of the vaccine. Immunization information was unknown for 9.4% (32/342) of the children overall. Routine immunization in South Africa is targeted at 6, 10 and 14 weeks of age, with rotavirus vaccine recommended at 6 and 14 weeks of age.

### 3.2. Rotavirus Infection in the Sample Population

Rotavirus was detected in 14.3% (49/342) of the diarrheal stool samples from children with diarrhea overall. The proportion of rotavirus-positive cases mirrored the age distribution of the study sample—thus, in children less than 12 months of age, encompassing 60.5% of the sample size (207/342), rotavirus was identified in 30 cases (61.2% of the study sample). Forty of the forty-nine rotavirus-positive cases were detected in children < 18 months of age (81.6%) (Table 1). Older children above 24 months of age were hardly infected. 

Only four children were documented to have not been immunized with rotavirus vaccine. None of these four children were positive for rotavirus diarrhea. The majority of children infected with rotavirus had received at least one dose of Rotarix vaccination (77.6% two doses; 10.2% single dose), although the vaccine record for six (12.2%) was unavailable. These six children were 6-, 12-, 13-, 18-, 37-, and 56- months old. In comparison, the vaccine record was not available for 26 (8.9%) of the 293-rotavirus negative children. Rotavirus cases were detected only in June–July 2015, July–August 2016, and June–September 2017, which are mid-winter and early spring months in South Africa. 

Other clinical symptoms experienced by the children with diarrhea included fever, vomiting, and refusal to eat. Of these symptoms, fever and vomiting were statistically significant to rotavirus infection (*p*-value of <0.05). To substantiate further, rotavirus infection was stratified by site as shown in Appendix A. Although more children overall (70.5%, 241/342) were admitted at OPHC with diarrheal disease, more children with rotavirus (59.2%; 29/49 versus 40.8%; 20/49) were admitted in DGMAH requiring hospitalization (Appendix A). Furthermore, the children admitted to the wards at DGMAH presented with fever, vomiting, and acute diarrhea of a shorter duration.

### 3.3. Rotavirus Infection and Circulating Rotavirus Strains

A diversity of rotavirus strains was detected in this study as shown in Figure 1. Overall, rotavirus G9P[8] (38.8%; 19/49) and G3P[4] (32.7%; 16/49) were the prevalent strains during the course of the study. Other strains detected included G9P[6] (6.1%; 3/49), G8P[4] (6.1%; 3/49), and G1P[8] (6.1%; 3/49). There were single detections of G2P[4], G3/G12P[4], G3/G9P[8], NEGP[8] and untypeable strains. A difference in genotype distribution throughout the study period was observed, rotavirus strain G9P[8] was dominant in 2015, while G3P[4] was predominant in 2016 and 2017. 

Interestingly, there was also a difference noted in the genotypes circulating in patients at DGMAH and OPHC. In DGMAH, rotavirus G3P[4] (48.3%; 13/29) was the most dominant strain detected, whereas at OPHC, G9P[8] (65%; 13/20) was the most dominant strain circulating. Other strains, such as G9P[6] and the single strains G2P[4], G3/G12P[4] and G3/G9P[8], were detected in DGMAH. 

### 3.4. Distribution of ABO Blood Groups, Lewis Antigens and Secretor Status in Rotavirus Infected Children

Based on the HBGA characterization noted above, 19 saliva samples were inconsistent in their classification and were excluded from analyses. Thus, a total of 323 samples were analyzed for both rotavirus and HBGA types. The distribution of blood groups in the population demonstrated that blood type O was most common in this population (62.5%; 202/323), followed by A (26.0%; 84/323), B (9.3%; 30/323), and then AB (2.2%; 7/323). As shown in Table 2, Pearson’s chi-squared test and Fisher’s exact tests revealed there is an association between rotavirus infection and the ABO groups, Lewis antigens, secretor status, and combined secretor and Lewis categories (all had p values less than 0.05). Children that were rotavirus infected were mostly blood group A (38.8%; 19/49) and blood group O (44.9%; 22/49). Blood group A was significantly associated with rotavirus indicated by an cOR of 2.39 95% CI of [1.21–4.71] and *p*-value of 0.011. 

In the current study, the population consisted of 82.7% (267/323) secretors and 17.3% (56/323) of non-secretors. From these 72.3% (146/202) and 27.3% (56/202) were blood group type O secretors and non-secretors, respectively. Among the rotavirus infected children, 93.9% (46/49) were secretors and 6.1% (3/49) non-secretors, highlighting the association of secretor status with rotavirus susceptibility (cOR 3.68 95% CI [1.28–15.55] 0.034). In comparison, of the rotavirus negative children, 80.7% (221/274) were secretors and 19.3% (53/274) were non-secretors, reflecting the population breakdown of secretor types. 

Furthermore, most children infected with rotavirus were secretors bearing the Lewis antigens Le^(a+b+)^, followed by Le^(a−b+)^; Lewis-negative children (^Le(a−b−)^ were the least infected. None of the Le^(a+b−)^ were infected with rotavirus. Multivariate logistic regression indicated there was no significant association between rotavirus infection to any of the HBGA predictors (Table 3).

### 3.5. Rotavirus Genotype Association to H1 Blood Group Antigens

Rotavirus P[4] strains (n = 21) were observed to only infect secretors, among which 57.1% (12/21) exhibited the Le^(a+b+)^ phenotype and 38.1% (8/21) had the Le^(a−b+)^ antigens. However, there was a single child with the Le^(a−b−)^ phenotype who was infected, and who exhibited the H type 1 antigen (i.e., A secretor). Children infected with rotavirus P[8] strains, exhibited all HBGA antigen phenotypes;91.7% (22/24) were secretors, 8.3% (2/24) non-secretor, 54.2% (13/24) Le^(a+b+)^, 20.8% (5/24) Le^(a−b+)^, and 25.0% (6/24) Le^(a−b−)^ (Table 4). 

P[6] strains infected three Lewis-negative (Le^(a−b−)^) children aged 5, 7, and 13 months. Two of the children were secretors possessing the blood type A and B while one was a non-secretor with blood type O.

## 4. Discussion

Globally, various studies have reported a putative association between the HBGAs and susceptibility to rotavirus infection, which may be driven by specific rotavirus strains as characterized by their VP4 genotype. Most have concluded that the presence of at least one of the genes (*FUT2* and *FUT3*) encoding HBGA has been associated with greater susceptibility to rotavirus infection [17,29,30]. This was confirmed in this study with greater susceptibility noted amongst the blood group A and O children. Furthermore, several studies have examined the relationship between rotavirus vaccine response and the observed lower efficacy in sub-Saharan Africa and South Asia in populations categorized by their HBGA [22,23,31,32,33,34]. Nevertheless, it remains unclear whether the HBGAs have a dramatic impact on rotavirus infection susceptibility or response to rotavirus vaccines. Recent data demonstrating specific binding domains within the H-1 type antigen and the P[8] strains may provide a mechanistic event by which this occurs [24], and this should be explored further in new studies. In the present study, we conducted the analysis of the H1-related blood group phenotypes in our local population to provide an early assessment of whether symptomatic rotavirus infection in South African children may be linked to blood group antigens.

South Africa was the first African country to introduce the rotavirus vaccine [35]. As observed globally, the introduction of the monovalent vaccine Rotarix had a great impact on the mortality and morbidity of rotavirus infection in South Africa [36,37]. During the period of this study, to examine the etiology of acute gastroenteritis disease in Northern Pretoria, South Africa post introduction (June 2015–December 2017), rotavirus accounted for 14.3% of diarrheal cases seeking medical attention or hospitalization. Most of the rotavirus positive children were hospitalized in DGMAH compared to outpatient treatment at OPHC. This may be an indication of more severe diarrheal disease caused by rotavirus requiring hospitalization and was corroborated by the observed higher rates of fever and vomiting in these hospitalized children. Interestingly, almost all rotavirus infected children had been vaccinated against rotavirus, having received at least one dose of Rotarix. Despite this, the effectiveness of the vaccine can be seen when comparing our findings to the 2003–2006 surveillance study pre-introduction, which reported 22.8% of rotavirus infections in children with gastroenteritis at DGMAH alone [38]. 

The introduction of the vaccine has had an impact not only on the reduction of rotavirus burden of disease, but also on circulating genotypes globally, including South Africa [39]. Characterization of the infecting genotypes revealed a diversity of strains, including those known to be commonly circulating globally such as G9P[8], G9P[6], G2P[4], and G1P[8], mixed strains (G3/G1, G3/G9), untypeable strains, and atypical strains such as G8P[4] and G3P[4]. Additionally, the genotype distribution displayed yearly fluctuations in this study, and this is a feature of rotavirus disease that has been reported previously in this region. Interestingly, the genotypes differed between the two sites which are located 23.6 km from each other. OPHC is an outpatient clinic and refers severe diarrhea cases requiring hospitalization to DGMAH, similar to other primary district hospitals and primary health care clinics in the catchment area of the tertiary care DGMAH. This wide catchment area which includes more rural areas which could have influenced the difference in the circulating genotypes at the two sites. Even so, year to year strain fluctuations, mixed infections, and untypeable and atypical rotavirus strains are common in Africa regardless of vaccination status of the population [28,40,41].

Rotaviruses are known to recognize HBGAs in a strain specific manner, suggesting that we may be able to map rotavirus genotype distribution according to HBGA profiles globally. If rotavirus infection is dependent on α1,2-linked fucose residue present in secretors as indicated in other studies [15,42], non-secretors should be somewhat resistant to some rotavirus strains and infection. In this study, the rotavirus infected children were mostly secretors (Lewis-positive secretors 79.6% and Lewis-negative secretors 14.3%) although this correlation was not absolute; 6.1% of Lewis-negative non-secretor children were infected by rotavirus. The secretor status may have played a role here.

Previous observational studies have reported conflicting results on the association between rotavirus and HBGA. Especially, the association of infecting rotavirus P-genotypes to the blood group antigens. For instance, in France [15], Nicaragua, and Burkina Faso [17], rotavirus P[8] exclusively infected secretors, which led to the conclusion that non-secretors may be resistant to rotavirus infection with P[8] strains. Recently, other studies have also observed P[8] infection in non-secretors, such as in South Africa [43], Spain [44], and Tunisia [21], although statistically, infections were more likely to be in secretors. Our study reports similar results in which P[8] infected both secretors and non-secretors but mostly secretors. Previously, rotavirus P[4] has been reported to share the same host preference with P[8] [9]. However, this was not observed in our study participants as P[4] infected almost exclusively Lewis-positive secretors; this has been reported elsewhere [45].

Furthermore, although the numbers were very low (n = 3), the association of P[6] strains with Lewis-negative subjects was noted, confirming previous results. Initially, it was suggested that rotavirus P[6] strains infect only Lewis-negative individuals, which was postulated as a reason for their prevalence among circulating strains in Africa where the Lewis-negative phenotype is common [9,29]. Unfortunately, we only identified three P[6] strains, and although we found that P[6] exclusively infected Lewis negative children in a population comprising of 11.2% Lewis negative profile, the numbers are too small to draw any real conclusion. Previously, P[6] rotaviruses were detected at approximately 9% in South Africa during the period 2006-2014 and were the third most common P-type [39]. This figure was 25% of circulating strains in northern Pretoria where this study was conducted, in 2003–2006 [38]. 

However, surveillance studies in other parts of the world have shown P[6] to infect Lewis-positive individuals too [17]. This was illustrated in in vitro binding assays as well, where P[6] weakly bound Lewis b antigens [16,45]. Altogether, this implies that P[4] and P[8] infection is significantly higher in Lewis-positive secretors and P[6] in Lewis-negative individuals, regardless of secretor status. 

Interestingly, none of the 45 children characterized as Le^(a+b−)^ were infected by rotavirus. It was previously shown in Nicaragua that children characterized as Le^(a+b−)^ did not seroconvert when vaccinated with either Rotarix or RotaTeq [31]. Similar results were also observed in South African children, 7 days post-vaccination [46]. The Le^(a+b−)^ phenotype lacks the fucose added by the FUT2 enzyme in the alpha 1,2 linkage which is present in secretors [47]. This proposes that the lack of alpha 1,2 fucose in this position is restrictive to vaccine replication [30] and may be protective of natural rotavirus infection.

Most children in this study (54.5%; 176/323) were phenotyped as Le^(a+b+)^ and a similar proportion (53.1%; 26/49) were shedding rotavirus. This HBGA phenotype is considered uncommon globally, although it has been reported in 51.9% of Black Brazilians of African descent [48], supporting that Le^(a+b+)^ may be common among African populations, as seen in our study.

Our study had few limitations. Firstly, relatively few cases of rotavirus diarrhea were detected as a result of the impact of rotavirus vaccination in South Africa which has significantly reduced diarrheal hospitalization and death [36,37]. In addition, the HBGA binding profiles of recombinant VP4/VP8* proteins of corresponding rotaviruses to saliva samples and genotyping of *FUT2* and *FUT3* genes of the saliva samples were not conducted. Previous studies that corroborated saliva binding assay results with genotyping showed compatibility within the results [31]. Disappointingly, the numbers of rotavirus genotypes with a VP4 P[6] were low (n = 3), which did not enable an evaluation of the putative association between P[6] and HBGAs as reported elsewhere.

## 5. Conclusions

In conclusion, this supplements the few studies conducted in African countries on the association of rotavirus to HBGAs. The present study shows that the presence (secretors) and absence (non-secretors) of HBGAs does not affect rotavirus infection but may impact the viral genotype diversity causing infection. Rotaviruses and HBGA are ubiquitous in different populations, although both show differences in frequency. Their interaction is complicated and even more so is their impact on geographical strain diversity and vaccine effectiveness. More studies are necessary to investigate the exact role of HBGAs as ligands for rotaviruses or even other ligands that could serve as rotavirus receptors.

## Figures and Tables

**Figure 1 viruses-14-02735-f001:**
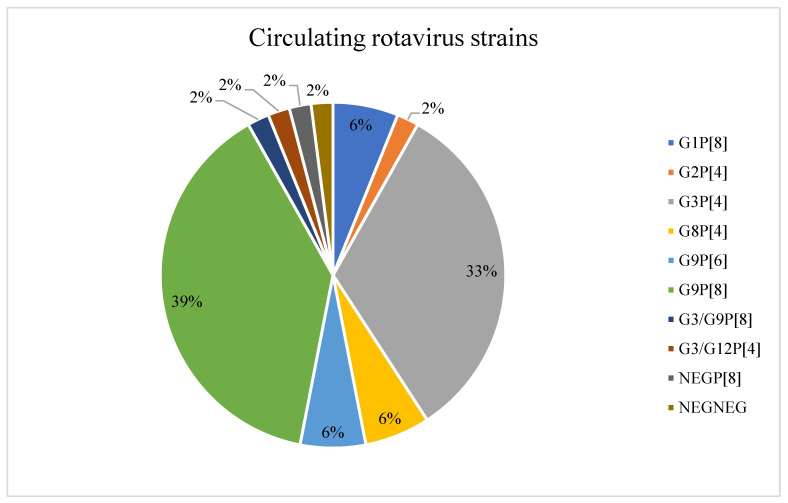
Circulating rotavirus strains in the children with gastroenteritis in northern Pretoria, South Africa in 2015-2017.

**Table 1 viruses-14-02735-t001:** Distribution of demographic data in rotavirus positive and rotavirus negative children.

	Total	RV Positive n (%) [95%CI]	RV Negative n (%) [95%CI]
Patients	342 (%)	49 (14.3%)	293 (85.7%)
**Sex**	*p*-value = 0.14 ^1^		
Male	176 (51.5%)	30 (61.2%) [46–74%]	146 (49.8%) [44–56%]
Female	166 (48.5%)	19 (38.8%) [26–54%]	147 (50.2%) [44–56%]
**Age of children in months**	*p*-value = 0.5 ^1^		
0–6	93 (27.2%)	10 (20.4%) [11–35%]	83 (28.3%) [23–34%]
7–12	114 (33.3%)	20 (40.8%) [27–56%]	94 (32.1%) [27–38%]
13–18	62 (18.1%)	10 (20.4%) [11–35%]	52 (17.7%) [14–23%]
19–24	36 (10.5%)	6 (12.2%) [5.1–25%]	30 (10.2%) [7.1–14%]
25–59	37 (10.8%)	3 (6.1%) [1.6–18%]	34 (11.6%) [8.3–16%]
**Rotavirus immunization**	*p*-value = 0.5 ^2^		
0 dose (unvaccinated)	4 (1.2%)	0 (0) [0.00–9.1%]	4 (1.4%) [0.44–3.7%]
1 dose	23 (6.7%)	5 (10.2%) [3.8–23%]	18 (6.14%) [3.8–9.7%]
2 doses	283 (82.7%)	38 (77.6%) [63–88%]	245 (83.6%) [79–88%]
No RTHC ^3^	32 (9.4%)	6 (12.2%) [5.1–25%]	26 (8.9%) [6.0–13%]
**Clinical symptoms ^4^**			
Fever (n = 334)	***p*-value = 0.005 ^1^**		
Yes	87 (26.0%)	20 (42.5%) [29–58%]	67 (23.3%) [19–29%]
None	247 (74.0%)	27 (57.4%) [42–71%]	220 (76.7%) [71–81%]
Vomiting (n = 332)	***p*-value < 0.001 ^1^**		
Yes	70 (21.1%)	21 (44.7%) [30–60%]	49 (17.2%) [13–22%]
None	262 (78.9%)	26 (55.3%) [40–70%]	236 (82.8%) [78–87%]
Refusal to eat (n = 332)	*p*-value = 0.11 ^1^		
Yes	108 (32.5%)	20 (42.5%) [29–58%]	88 (30.9%) [26–37%]
None	224 (67.5%)	27 (57.5%) [63–74%]	197 (69.1%) [42–71%]
Duration of diarrhea days	4.22	3.27	4.32

^1^ Pearson’s chi-squared test. ^2^ Fisher’s exact test. ^3^ Road to Health Card. ^4^ All the children had diarrhea.

**Table 2 viruses-14-02735-t002:** Summary of Secretor Status and Lewis antigen phenotypes in rotavirus positive and negative children.

Phenotype	Total (%)	RV Positive (%)	RV Negative (%)
**Patient sample**	323	49	274
**ABO groups**	***p*-value = 0.002 ^2^**		
A	84 (26%)	19 (38.8%)	65 (23.7%)
AB	7 (2.2%)	4 (8.2%)	3 (1.1%)
B	30 (9.3%)	4 (8.2%)	26 (9.5%)
O	202 (62.5%)	22 (44.9%)	180 (65.7%)
**Lewis phenotype**	***p*-value = 0.004 ^1^**		
Le^(a−b−)^	36 (11.2%)	10 (20.4%)	26 (9.5%)
Le^(a+b+)^	176 (54.5%)	26 (53.1%)	150 (54.8%)
Le^(a−b+)^	66 (20.4%)	13 (26.5%)	53 (19.3%)
Le^(a+b−)^	45 (13.9%)	0 (0.00%)	45 (16.4%)
**Secretor status**	***p*-value = 0.024 ^1^**		
Non-secretor	56 (17.3%)	3 (6.1%)	53 (19.3%)
Secretor	267 (82.7%)	46 (93.9%)	221 (80.7%)
**Combined**	***p*-value < 0.001 ^2^**		
Sec/le-(Le^(a−b−)^)	25 (7.7%)	7 (14.3%)	18 (6.6%)
Sec/Le+ (Le^(a−b+)^; Le^(a+b+)^)	242 (74.9%)	39 (79.6%)	203 (74.1%)
Non-sec/Le+ (Le^(a+b−)^)	45 (13.9%)	0 (0)	45 (16.4%)
Non-sec/le-(Le^(a−b−)^)	11 (3.4%)	3 (6.1%)	8 (2.9%)

^1^ Pearson’s chi-squared test. ^2^ Fisher’s exact test.

**Table 3 viruses-14-02735-t003:** Simple and multivariate logistic regression of ABO groups, Lewis antigens, and secretor status association with rotavirus infection.

Logistic Regression	cOR (95%CI)	*p*-Value (Wald’s Test)	aOR (95%CI)	*p*-Value
**ABO groups vs. RV infection**				
Ref O				
A	2.39 [1.21–4.71]	0.011	1.84 [0.89–3.77]	0.097
AB	10.9 [2.27–58.49]	0.003	8.99 [1.83–48.96]	0.006
B	1.26 [0.35–3.62]	0.693	0.92 [0.25–2.77]	0.892
**Lewis antigen vs. RV infection**				
Ref Le^(a+b+)^				
Le^(a−b−)^	2.22 [0.93–5.05]	0.063	2.23 [0.78–5.89]	0.117
Le^(a−b+)^	1.42 [0.66–2.91	0.355	1.36 [0.62–2.87]	0.423
Le^(a+b−)^	0.00 [0.00–infinite]	0.986	0.00 [0.00–infinite]	0.986
**Secretor status vs. RV infection**				
Ref Non-secretor				
Secretor	3.68 [1.28–15.55]	0.034	0.77 [0.15–4.61]	0.762

**Table 4 viruses-14-02735-t004:** Secretor and Lewis phenotypes and infecting rotavirus genotypes.

	Rotavirus P-Genotypes		
**Phenotypes**	P[8] 24 (%)	P[4] 21 (%)	P[6] 3 (%)
**ABO groups**			
A	10 (41.7%)	7 (33.3%)	1 (33.3%)
AB	1 (4.2%)	3 (14.3%)	-
B	2 (8.3%)	1 (4.8%)	1 (33.3%)
O	11 (45.8%)	10 (47.6%)	1 (33.3%)
**Lewis phenotype**			
Le^(a+b+)^	13 (54.2%)	12 (57.1%)	-
Le^(a−b+)^	5 (20.8%)	8 (38.1%)	-
Le^(a−b−)^	6 (25%)	1 (4.8%)	3 (100%)
**Secretor status**			
Non-secretor	2 (8.3%)	-	1 (33.3%)
Secretor	22 (91.7%)	21 (100.0%)	2 (66.7%)
**Combined**			
Sec/Le+ (Le^(a−b+)^; Le^(a+b+)^)	18 (75%)	20 (95.2%)	-
Sec/le-(Le^(a−b−)^)	4 (16.7%)	1 (4.8%)	2 (66.7%)
Non-sec/le-(Le^(a−b−)^)	2 (8.3%)		1 (33.3%)

## Data Availability

Not applicable.

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
