# Peer review of "The Association between Symptomatic Rotavirus Infection and Histo-Blood Group Antigens in Young Children with Diarrhea in Pretoria, South Africa"

_viruses, 2022, doi:10.3390/v14122735_

Round 1
Reviewer 1 Report
Comments to the authors
1) Title: suggest removing “suffering” and just state “in young children with diarrhoea in Pretoria…”. Revise throughout as “suffering” is not a well-defined word in a medical context.
2) Abstract:
Line 31-32: “Blood type A children were significantly more likely to be infected by rotavirus (38.8%) than any other blood types, followed by AB types (8.2%).” – level of significance e.g. p-value is not given to substantiate this statement.
Line 32-33: “P[4] rotaviruses (n=21) infected only secretor individuals, whereas P[6] rotaviruses (n=3) only infected Le(a-b-).” Provide the total number of rotavirus positives and percent that were P[4] and P[6].
Line 35-36 “Our findings demonstrated that Lewis antigens serve as susceptibility factors to the P[6] rotaviruses to facilitate viral infection – unclear what this means as the P[6] only infected Lewis negatives. In addition, the sample size for P[6] is very small (n=3).
3) Study population: This needs to be described in more detail. How were the children enrolled? What were the eligibility criteria, inclusion/exclusion criteria? How was acute gastroenteritis defined? Moderate or severe or any diarrhoea? Duration? Age of children? Type of sampling? How was sample size determined, including sample size for each site? How was vaccination status obtained?
4) Demographics: this should be in the results section not the methods.
5) Sample collection: was this done on day of enrolment? Were stool samples tested in real time or stored? Provide details.
6) HBGA phenotyping needs to be described in more detail. How were the HBGA categories defined? What were the Elisa cutoffs? Were there any samples that could not be classified?
7) Statistical analysis needs clarification and additional details. What was the primary outcome? The comparison seems to be between HBGA and rotavirus positivity but this needs to be clearly stated and the exact methods used need to be clarified. There is no mention of comparison between sites in the methods yet Table 1 is stratified by site. Are the p-values and OR in Table 1 for the comparison between sites? How was the OR calculated – logistic regression? Was there adjustment for confounding? Why was Fishers exact test used to calculate the p-values? The OR in table 1 and 2 are difficult to interpret as the comparison is unclear and there is no reference group for the categorical variables. Please clarify.
8) Results:
· Line 181: give proportion of rotavirus in the text.
· If stating that something is significant then a p-value/OR must be provided to substantiate the statement e.g. line 185-6
· Only put results in the results section. If suggesting reasons for the results, this should be in the discussion e.g. line 189-191.
· Line 194-195 “The vaccine record for six (12.25%) of the 49 194 rotavirus-infected children was unavailable.” How did this compare to the rotavirus-negative children - similar proportion?
· Table 1: in the methods it seems that the primary comparison is between rotavirus positive and negative yet here the comparison is between sites. Please clarify the main objective of the study and provide the results accordingly. If there are multiple comparisons, then these need to be detailed in the methods section. It’s unclear which is the reference category for the categorical variables e.g. age group and vaccination status, thus the OR are difficult to interpret. E.g. OR for vaccination status of 1.79 – it’s unclear what this OR is referring to: at least one dose compared to no vaccination? There are so few rotavirus positives in the older age groups - suggest making into a single group e.g. 24-60 months.
· Line 237-238: “Of the rotavirus infected children, 85.7% 237 (42/49) were secretors and 14.3% (7/49) non-secretors, this association was not significant.” What association is being referred to here?
· Table 2: once again, there does not seem to be a reference group for the categorical variables e.g. blood group. What is the OR referring to?
· How can you explain the very high proportion of Le(a+b+)? This is not consistent with the literature and this is not discussed at all in the discussion.
· Le(a+b-) is not consistent with secretor positive status yet this appears in the combined classification in Table 2. Please clarify.
9) It is very difficult to comment on the discussion without clarification of the methods and results.

Author Response
Responses to Reviewer 1:
Title: suggest removing “suffering” and just state “in young children with diarrhoea in Pretoria…”. Revise throughout as “suffering” is not a well-defined word in a medical context.
Response: We have removed the word suffering from the Title and elsewhere in the manuscript.
2) Abstract:
Line 31-32: “Blood type A children were significantly more likely to be infected by rotavirus (38.8%) than any other blood types, followed by AB types (8.2%).” – level of significance e.g. p-value is not given to substantiate this statement.
Response: p-values have been added to Table 2 and the Results section to substantiate this statement.
Line 32-33: “P[4] rotaviruses (n=21) infected only secretor individuals, whereas P[6] rotaviruses (n=3) only infected Le(a-b-).” Provide the total number of rotavirus positives and percent that were P[4] and P[6].
Response: Percentages of P[4] and P[6] were added in lines 32-33
Line 35-36 “Our findings demonstrated that Lewis antigens serve as susceptibility factors to the P[6] rotaviruses to facilitate viral infection – unclear what this means as the P[6] only infected Lewis negatives. In addition, the sample size for P[6] is very small (n=3).
Response: We have revised this text to say “Our findings demonstrated that Lewis antigens, or the lack thereof, may serve as susceptibility factors to rotaviral infection by specific VP4 genotypes as observed elsewhere. Potentially, the P[8] strains remain the predominant human VP4 genotype due to their ability to bind to a variety of HBGA phenotypes.
Study population: This needs to be described in more detail. How were the children enrolled? What were the eligibility criteria, inclusion/exclusion criteria? How was acute gastroenteritis defined? Moderate or severe or any diarrhoea? Duration? Age of children? Type of sampling? How was sample size determined, including sample size for each site? How was vaccination status obtained?
Response: More information on the study population has been added in lines 101-120. In essence, this study was embedded within a national diarrhea surveillance program that has been ongoing for years, to assess the impact of rotavirus vaccine introduction and to identify the etiological causes of acute diarrhea disease in young children. As such, all diarrhea cases presenting to Dr. George Mukhari Academic Hospital or surrounding clinics were identified and a stool sample captured. Parents were offered participation in the study and after giving informed consent, the children were enrolled in this study.
Demographics: this should be in the results section not the methods.
Response: We have moved this section to the Results section.
Sample collection: was this done on day of enrolment? Were stool samples tested in real time or stored? Provide details.
Response: We revised the text on 131-138 to address this comment.
HBGA phenotyping needs to be described in more detail. How were the HBGA categories defined? What were the Elisa cutoffs? Were there any samples that could not be classified?
Response: The HBGA phenotyping method has been rewritten with more detail, including information of the signal cutoff (lines 161-174). Based on HBGA biosynthesis pathway, different HBGA categories are defined as following:
1) secretors are defined as those who have one or more H1 antigen-related HBGA types, including A, B, H1, and/or Leb antigen;
2) non-secretors are defined as those who lack the H1 antigen-related HBGA types and are positive with Lea antigen;
3) Le(a-b+) individuals are Lea negative but Leb positive;
4) Le(a+b+) are those with both Lea and Leb antigens;
5) Le(a+b-) are those with Lea positive, but Leb negative, and
6) Le(a-b-) individuals are those lacking both Lea and Leb antigens.
This information has also been added to the HBGA phenotyping section.
19 samples could not be classified and thus were excluded from analysis of the study. This was added to the Results section (line 274).
Statistical analysis needs clarification and additional details. What was the primary outcome? The comparison seems to be between HBGA and rotavirus positivity but this needs to be clearly stated and the exact methods used need to be clarified. There is no mention of comparison between sites in the methods yet Table 1 is stratified by site. Are the p-values and OR in Table 1 for the comparison between sites? How was the OR calculated – logistic regression? Was there adjustment for confounding? Why was Fishers exact test used to calculate the p-values? The OR in table 1 and 2 are difficult to interpret as the comparison is unclear and there is no reference group for the categorical variables. Please clarify.
Response: Thank you for this comment. The statistical analysis has been amended in the manuscript and additional information added (lines 188-194).
Results:
Line 181: give proportion of rotavirus in the text.
Response: The proportion of rotavirus infected children has been added (lines 222-onwards).
If stating that something is significant then a p-value/OR must be provided to substantiate the statement e.g. line 185-62
Response: The Results section has been substantially re-written and statistical results added to substantiate any significant findings.
Only put results in the results section. If suggesting reasons for the results, this should be in the discussion e.g. line 189-191.
Response: We have removed this commentary and placed it in the Discussion.
Line 194-195 “The vaccine record for six (12.25%) of the 49 194 rotavirus-infected children was unavailable.” How did this compare to the rotavirus-negative children - similar proportion?
Response: We have added a sentence to indicate that the vaccine record was not available for 26 (8.9%) of the 293 rotavirus negative children - added in lines 214-216.
Table 1: in the methods it seems that the primary comparison is between rotavirus positive and negative yet here the comparison is between sites. Please clarify the main objective of the study and provide the results accordingly. If there are multiple comparisons, then these need to be detailed in the methods section. It’s unclear which is the reference category for the categorical variables e.g. age group and vaccination status, thus the OR are difficult to interpret. E.g. OR for vaccination status of 1.79 – it’s unclear what this OR is referring to: at least one dose compared to no vaccination? There are so few rotavirus positives in the older age groups - suggest making into a single group e.g. 24-60 months.
Response: Thank you for the comment, the comparison was indeed between rotavirus positive and negative cases. The additional breakdown of results based on sites has been included as supplementary table in the revised version of the manuscript to substantiate our findings. The statistical analysis section has been amended as well.
Line 237-238: “Of the rotavirus infected children, 85.7% 237 (42/49) were secretors and 14.3% (7/49) non-secretors, this association was not significant.” What association is being referred to here?
Response: We have added clarity here to highlight that we mean the association between rotavirus infection and the secretor status of the individual (lines 287-289).
Table 2: once again, there does not seem to be a reference group for the categorical variables e.g. blood group. What is the OR referring to?
Response: The logistic regression modelling was conducted to predict rotavirus with HBGA category and reference groups are indicated in Table 3.
How can you explain the very high proportion of Le(a+b+)? This is not consistent with the literature and this is not discussed at all in the discussion.
Response: We note this difference, which has now been discussed under the discussion section (lines 411-416).
Le(a+b-) is not consistent with secretor positive status yet this appears in the combined classification in Table 2. Please clarify.
Response: We thank the reviewer for pointing this out. Table 2 has been amended to remove the inconsistency.
It is very difficult to comment on the discussion without clarification of the methods and results.
Response: We hope that the clarifications that we have included in this revised version, will offer greater opportunity for the reviewer to review the discussion of our results.
Reviewer 2 Report
The manuscript by Rakay et al. investigates the association between HBGAs and susceptibility to rotavirus infection in symptomatic children in South Africa.The association between HBGAs and rotavirus was observed relatively recently, and there are some conflicting results between studies in different countries. As such, the theme is of importance to understand susceptibility, epidemiology and putatively also effect on protection after rotavirus vaccination; and the manuscript can thus provide valuable complementary knowledge to the field.
Nevertheless, there are some methodological/classification issues with regards to phenotyping/classification of the saliva phenotypes that needs to be addressed in a revised version.
Major:
There are some inconsistencies with the phenotyping/methodology; particularly the definition of secretor/Lewis phenotypes in saliva that needs to be better clarified before publication as this is essential for the results and interpretations presented here.
Methods and results with regards to HBGA phenotyping
line 153. HBGA phenotyping in saliva. The authors should here clearly state how they defined secretor positive or secretor negative phenotype: whether inconsistencies were observed (e.g. between Lewis, secretor, ABO), and how these inconsistencies were addressed/classified. As these HBGAs belong to the same biosynthetic pathway, it can be relatively easy to find whether the results are consistent; e.g. a non-secretor should always be typed as ABO type O; and should be Lea+b- (if Lewis positive); and a secretor should express Leb (if Lewis positive).
e.g. Table 3. P[8] infections; n=24. P[8] infected 5 non-secretors and 19 secretors. But P[8] were also found in 13 Lea+b-. Lea+b- is a phenotype of non-secretors (as the authors also state on page 7 “…non-secretors defined by Lea+b-, Lea-b- H type 1 negative”, and in introduction “…..leaving only Le(a) positive phenotypes, which are defined as non-secretor types” as well as in discussion. Does this mean that H type 1 positive samples were phenotyped as Lea+b-?. How to explain this discrepancy (secretors should express Leb)? Also in discussion “Interestingly none of the 64 children characterized as Lea+b- were infected by rotavirus”. Are these not non-secretors? Saliva phenotyping that yielded discrepant result; could be tested with other methods for secretor phenotyping, example by UEA-1 lectin to ensure correct classification.
ABO: As the blood grouping was made on saliva (as I understand from manuscript) then it is only possible to determine ABO blood group in secretors (as non-secretors per definition will not express these). Thus non-secretors will be incorrectly classified as blood group O if measuring saliva. Suggest clarifying this; and also make the ABO blood group analysis only between secretors; or alternatively have an additionally category called non-secretor.
Other
Abstract first line suggest "receptors or attachment factors"
Abstract last line “that Lewis antigens serve as susceptibility factors”. I understand what the authors mean; but is it not rather Lewis-negative phenotype that is a susceptibility factor for P[6]?
Introduction “On the other hand, FUT3 gene encodes the α1,3/4 fucosyltransferase (FUT3) enzyme that facilitates the production of Lewis antigens, including Lewis a (Le(a+)) and Lewis b (Le(b+)), by adding an α1,3/4 fucose to the type I precursor”. Some of the information here is incorrect. FUT3 can also add fucose on the H1 antigen making Leb (not only on the type I precursor). Also when FUT3 works on the type I HBGA, it will be a α1,4 linkage; not α1,3/4.
“in most cases, FUT2 gene is normal and FUT2 enzyme is active”
Would suggest to rephrase and specify approx.. % of secretor positive phenotype, globally and/or regionally”
“Rotavirus P[6] strains are reported to occur at a higher prevalence in non-secretors [9]”
Is this the correct reference, it is an invitro study? I believe most studies have reported that P[6] infections predominantly occur in Lewis-negatives; independently of secretor status. Suggest checking this and revise accordingly
“It has also been postulated that the higher circulation of P[6] strains in Africa is linked to this same population factor”. Unclear what population factor refers to? I believe P[6] is linked to Lewis-negative phenotype which is common in sub-Saharan Africa compared to most other countries where studies have been performed.
Results:
Could some of the G1P[8] strains have been from the vaccination, rather than wildtype infections? (e.g. detected early after vaccination, or similar).
Table 1. It is a bit unclear (to me) which comparison the p-value reflects? Which groups have been compared with which test?
Table 2. as above; it is strange to see the combination Se/Le+ include Lea+b-; as this is the Lewis phenotype associated with non-secretors.
Discussion: The fact the most children were vaccinated; could this have influenced the observed association between HBGA and rotavirus; compared to other studies with non-vaccinated children?
“Recently, other studies have also observed P[8] infection in non-secretors, such as in South Africa [42], Spain [43] and Tunisia [21}, which was similar to our findings” I believe this is incorrect as stated; the studies from Spain and South Africa observed, while P[8] infected non-secretors, that infections were statistically much more likely to occur in secretors; which is not the case with this study.
Line 314-315 “The children infected with P[8] were mostly Lewis positive secretors (75%) and Lewis negative non-secretors (20.8%)”. It is a bit hard to draw any conclusions from this statment as it depends on the distribution of HBGA in the general population; suggest rephrase or remove.
“Rotavirus P[4] shares the same host preference with P[8]; infecting almost exclusively Lewis positive secretors” This statement does not seem to reflect the results from this study (e.g. see Table 3); P[4] only found in secretors, whereas P[8] is not. Similar to findings in https://doi.org/10.1093/infdis/jiy054
Line 317 “Another disparity….” Suggest rephrase; as the results of this study is in line with previous, i.e. that P[6] predominantly infects Lewis-negatives; although numbers were low here.
The authors could also discuss findings from this study “Host-Range Shift Between Emerging P[8]-4 Rotavirus and Common P[8] and P[4] Strains”; showing that secretor specificity may also depend on the lineage of P[8]. It would be of interest if the authors could sequence the P[8] strains to determine this.
Author Response
Responses to Reviewer 2:
The manuscript by Rakay et al. investigates the association between HBGAs and susceptibility to rotavirus infection in symptomatic children in South Africa. The association between HBGAs and rotavirus was observed relatively recently, and there are some conflicting results between studies in different countries. As such, the theme is of importance to understand susceptibility, epidemiology and putatively also effect on protection after rotavirus vaccination; and the manuscript can thus provide valuable complementary knowledge to the field.
Nevertheless, there are some methodological/classification issues with regards to phenotyping/classification of the saliva phenotypes that needs to be addressed in a revised version.
Major:
There are some inconsistencies with the phenotyping/methodology; particularly the definition of secretor/Lewis phenotypes in saliva that needs to be better clarified before publication as this is essential for the results and interpretations presented here.
Methods and results with regards to HBGA phenotyping
line 153. HBGA phenotyping in saliva. The authors should here clearly state how they defined secretor positive or secretor negative phenotype: whether inconsistencies were observed (e.g. between Lewis, secretor, ABO), and how these inconsistencies were addressed/classified. As these HBGAs belong to the same biosynthetic pathway, it can be relatively easy to find whether the results are consistent; e.g. a non-secretor should always be typed as ABO type O; and should be Lea+b- (if Lewis positive); and a secretor should express Leb (if Lewis positive).
Response: We thank the reviewer for the constructive comments. As explained above in response to point #5, the HBGA phenotyping section has been rewritten with clear definitions of different HBGA categories used in this study. Based on these definitions, we have corrected the inconsistencies. We confirm that, as mentioned by the reviewer, a non-secretor should always be typed as type O in the ABO blood type system; and should be Lea+b- (if Lewis positive) in the Lewis system; and a secretor should express Leb (if Lewis positive).
e.g. Table 3. P[8] infections; n=24. P[8] infected 5 non-secretors and 19 secretors. But P[8] were also found in 13 Lea+b-. Lea+b- is a phenotype of non-secretors (as the authors also state on page 7 “…non-secretors defined by Lea+b-, Lea-b- H type 1 negative”, and in introduction “…..leaving only Le(a) positive phenotypes, which are defined as non-secretor types” as well as in discussion. Does this mean that H type 1 positive samples were phenotyped as Lea+b-?. How to explain this discrepancy (secretors should express Leb)? Also in discussion “Interestingly none of the 64 children characterized as Lea+b- were infected by rotavirus”. Are these not non-secretors? Saliva phenotyping that yielded discrepant result; could be tested with other methods for secretor phenotyping, example by UEA-1 lectin to ensure correct classification.
Response: We apologize for these errors and confusions. These data have been re-examined and classified to remove the mentioned inconsistencies. As mentioned above, the definitions of each HBGA categories have been clearly given. Regarding the HBGA phenotyping methods, the saliva-based ELISA using specific monoclonal antibodies is a widely accepted one that has been used in numerous other publications. Currently, we do not have the resources or laboratory methodologies to conduct further validation of these results through sequencing etc.
ABO: As the blood grouping was made on saliva (as I understand from manuscript) then it is only possible to determine ABO blood group in secretors (as non-secretors per definition will not express these). Thus, non-secretors will be incorrectly classified as blood group O if measuring saliva. Suggest clarifying this; and also make the ABO blood group analysis only between secretors; or alternatively have an additionally category called non-secretor.
Response: We wish to clarify that we determine the HBGA types of A, B, H1, Lea, and Leb by corresponding monoclonal antibodies. Based on the definitions described above, A, B, and AB blood types are those with A, B, and AB antigens, respectively. Type O individuals are those lacking both A and B antigens. They can be secretors if they are H1 positive, or non-secretors if they are H1 negative (Lea positive). Thus, blood types A and B were classified as secretors, while all the non-secretors were blood group type O as shown in the amended Table 2. We have added a sentence stratifying blood group O in the secretors and non-secretors in line 302-303.
Other
Abstract first line suggest "receptors or attachment factors"
Response: We have revised the text accordingly.
Abstract last line “that Lewis antigens serve as susceptibility factors”. I understand what the authors mean; but is it not rather Lewis-negative phenotype that is a susceptibility factor for P[6]?
Response: We have revised the text to say that “Lewis antigens, or the lack thereof, may serve as susceptibility factors to rotavirus infection”.
Introduction “On the other hand, FUT3 gene encodes the α1,3/4 fucosyltransferase (FUT3) enzyme that facilitates the production of Lewis antigens, including Lewis a (Le(a+)) and Lewis b (Le(b+)), by adding an α1,3/4 fucose to the type I precursor”. Some of the information here is incorrect. FUT3 can also add fucose on the H1 antigen making Leb (not only on the type I precursor). Also when FUT3 works on the type I HBGA, it will be a α1,4 linkage; not α1,3/4.
Response: Thank you. We have revised the highlighted sentences accordingly to say “On the other hand, FUT3 gene encodes the α1,3/4 fucosyltransferase (FUT3) enzyme that facilitates the production of Lewis antigens, including Lewis a (Le(a+)) and Lewis b (Le(b+)), by adding a fucose to the α1,4 linkage of the type I precursor and/or H type 1 antigen, respectively”.
“in most cases, FUT2 gene is normal and FUT2 enzyme is active” Would suggest to rephrase and specify approx.. % of secretor positive phenotype, globally and/or regionally”
Response: We thank the reviewer for this constructive suggestion and the sentence has been modified accordingly to say “In approximately 80% of the global population, FUT2 gene is functional and FUT2 enzyme is active, leading to production of H-related antigens, including H type 1, and/or Le(b+) and such HBGA phenotypes are called secretor types”.
“Rotavirus P[6] strains are reported to occur at a higher prevalence in non-secretors [9]”
Is this the correct reference, it is an invitro study? I believe most studies have reported that P[6] infections predominantly occur in Lewis-negatives; independently of secretor status. Suggest checking this and revise accordingly
Response: Thank you, we have clarified the text to be accurate. It should be reference #17
“It has also been postulated that the higher circulation of P[6] strains in Africa is linked to this same population factor”. Unclear what population factor refers to? I believe P[6] is linked to Lewis-negative phenotype which is common in sub-Saharan Africa compared to most other countries where studies have been performed.
Response: Several reports have indicated that the high prevalence of rotavirus P[6] strains, previously observed in Africa and circulating at a higher proportion than seen in other regions, may be due to the high proportion of the indigenous population who have Lewis-negative status. We have clarified the text in lines 92-94
Results:
Could some of the G1P[8] strains have been from the vaccination, rather than wildtype infections? (e.g. detected early after vaccination, or similar).
Response: We were able to confirm in most children that the diarrhea cases overall and the rotavirus cases specifically, were not associated with shedding of the rotavirus vaccine. This was not possible for the 6 children with no RTHC available, but they were 6-, 12-, 13- 18- 37- and 56-months old, and much older than the immunization schedule in South Africa. None of the rotavirus-positive children were enrolled within 4 weeks of immunization.
Table 1. It is a bit unclear (to me) which comparison the p-value reflects? Which groups have been compared with which test?
Response: Table 1 has been edited to make this clear. Each demographic (sex/age group/ vaccine doses/clinical symptom) data was compared in RV positive vs RV negative using either Chi-squared or Fishers exact test where appropriate.
Table 2. as above; it is strange to see the combination Se/Le+ include Lea+b-; as this is the Lewis phenotype associated with non-secretors.
Response: We thanks the reviewer for pointing this out. Table 2 has been revised to remove this confusion based on the definitions of the HBGA categories.
Discussion: The fact the most children were vaccinated; could this have influenced the observed association between HBGA and rotavirus; compared to other studies with non-vaccinated children?
Response: This is an interesting observation and has not been addressed in this study. Other studies have not documented a change in susceptibility to rotavirus infection based on immunization status (Armah et al, 2019; Pollock et al, 2018; Lee et al, 2018).
“Recently, other studies have also observed P[8] infection in non-secretors, such as in South Africa [42], Spain [43] and Tunisia [21}, which was similar to our findings” I believe this is incorrect as stated; the studies from Spain and South Africa observed, while P[8] infected non-secretors, that infections were statistically much more likely to occur in secretors; which is not the case with this study.
Response: Thank you for pointing out this discrepancy. We have revised the text in line with the earlier revisions mentioned above (lines 381-386).
Line 314-315 “The children infected with P[8] were mostly Lewis positive secretors (75%) and Lewis negative non-secretors (20.8%)”. It is a bit hard to draw any conclusions from this statement as it depends on the distribution of HBGA in the general population; suggest rephrase or remove.
Response: We have removed this statement.
“Rotavirus P[4] shares the same host preference with P[8]; infecting almost exclusively Lewis positive secretors” This statement does not seem to reflect the results from this study (e.g. see Table 3); P[4] only found in secretors, whereas P[8] is not. Similar to findings in https://doi.org/10.1093/infdis/jiy054
Response: This is indeed true and the statement has been amended to highlight the difference in host preference between P[8] and P[4] as seen in our study as follows.
“In this study, the rotavirus infected children were mostly Lewis positive secretors (93.9%) although this correlation was not absolute – 6.1% of Lewis negative children were infected by rotavirus.”
Line 317 “Another disparity….” Suggest rephrase; as the results of this study is in line with previous, i.e. that P[6] predominantly infects Lewis-negatives; although numbers were low here.
Response: We have revised the sentence to say
“Furthermore, although the numbers were very low (n=3), the association of P[6] strains was observed with Lewis negative subjects was noted, confirming previous results”.
The authors could also discuss findings from this study “Host-Range Shift Between Emerging P[8]-4 Rotavirus and Common P[8] and P[4] Strains”; showing that secretor specificity may also depend on the lineage of P[8]. It would be of interest if the authors could sequence the P[8] strains to determine this.
Response: Thank you for this comment and it is a line of enquiry that we would like to follow. However, it is not in scope for this particular project. We have however, added some text to state this possibility.
Reviewer 3 Report
This is an interesting manuscript that reports the association between symptomatic rotavirus infected young children and histo-blood group antigens in Pretoria, South Africa.
The main strengths of the article are the clearness of the methods, results and discussion as well as the adequate number of tables. On the other hand, there are main aspects that must be improved like:
Table 2 lacks data from healthy population to use these as control. The table was only elaborated with data from recruited children suffering from rotavirus gastroenteritis, so the conclusions obtained are relative to study population. To obtain global conclusions, the data must be extrapolated to the South Africa population. Authors should use South Africa population general data relative to ABO groups, Secretor status and Lewis phenotype to use as control group and perform a chi-squared distribution analysis to confirm their findings.
Regarding the literature available about the relation between the P[8] genotype rotavirus infection and secretor status (lines 82-87) the reviewer encourage authors to read the article from Gozalbo-Rovira et al (doi: 10.1371/journal.ppat.1007865) where the mechanistic under the differences in infection susceptibility between secretors and non-secretors in G1P[8] rotavirus were explained. This information should be also taken in consideration in lines 301-316 from the discussion section, where the authors use outdated data in line 304 (years 2012 and 2014).
As authors indicate in the line 111, the vaccine approved in South Africa is the Rotarix vaccine, authors should take in consideration for further discussion that this vaccine belongs to G1P[8] genotype and could give an explanation for the rotavirus strains identified in the present manuscript (lines 207-217).
Minor comments:
Thousands should be separated by comma in line 45.
In line 177, the number of children was mismatched from 342 to 323, the correct percentage is 74.6%.
In line 207 use were instead was.
There is a mismatch between the number of children with blood group A in line 228 (99/342) and the indicated in the table 2 (91). Related to this, the percentage for blood group B indicated in same line (10.5%; 33/342) is wrong, the correct is 9.7%.
The blood groups more frequent in infected children were A and O group, not AB that was indicated in line 230.
Author Response
Reviewer 3:
This is an interesting manuscript that reports the association between symptomatic rotavirus infected young children and histo-blood group antigens in Pretoria, South Africa.
The main strengths of the article are the clearness of the methods, results and discussion as well as the adequate number of tables. On the other hand, there are main aspects that must be improved like:
Table 2 lacks data from healthy population to use these as control. The table was only elaborated with data from recruited children suffering from rotavirus gastroenteritis, so the conclusions obtained are relative to study population. To obtain global conclusions, the data must be extrapolated to the South Africa population. Authors should use South Africa population general data relative to ABO groups, Secretor status and Lewis phenotype to use as control group and perform a chi-squared distribution analysis to confirm their findings.
Response: We agree that a healthy control group would have been beneficial, however the study recruited young children from the communities in Ga-Rankuwa Township who presented with diarrheal illness. There is no reason to consider that the children with diarrhea differ significantly from healthy, non-diarrheal children in this community. Our study examined the association of ABO blood groups, secretor status and Lewis phenotype in children with rotavirus diarrhea or non-rotavirus diarrhea.
Regarding the literature available about the relation between the P[8] genotype rotavirus infection and secretor status (lines 82-87) the reviewer encourage authors to read the article from Gozalbo-Rovira et al (doi: 10.1371/journal.ppat.1007865) where the mechanistic under the differences in infection susceptibility between secretors and non-secretors in G1P[8] rotavirus were explained. This information should be also taken in consideration in lines 301-316 from the discussion section, where the authors use outdated data in line 304 (years 2012 and 2014).
Response: Thank you for pointing to this recent data. We have added this in the Introduction (lines 93-95) and refer to it in the Discussion (lines 321-323). We have also added the reference.
As authors indicate in the line 111, the vaccine approved in South Africa is the Rotarix vaccine, authors should take in consideration for further discussion that this vaccine belongs to G1P[8] genotype and could give an explanation for the rotavirus strains identified in the present manuscript (lines 207-217).
Response: None of the rotavirus positive children were close to their immunization date.
Minor comments:
Thousands should be separated by comma in line 45.
Response: We have revised this as recommended.
In line 177, the number of children was mismatched from 342 to 323, the correct percentage is 74.6%.
Response: This error was corrected
In line 207 use’ were’ instead ‘was’.
Response: The verb "was" describes the ‘diversity’ so is singular. We have kept the original text.
There is a mismatch between the number of children with blood group A in line 228 (99/342) and the indicated in the table 2 (91). Related to this, the percentage for blood group B indicated in same line (10.5%; 33/342) is wrong, the correct is 9.7%.
Response: The error has been corrected- thank you
The blood groups more frequent in infected children were A and O group, not AB that was indicated in line 230.
Response: This sentence was revised. Statistically AB indicated a significant odd ratio to rotavirus however the 95% CI was to wide.
Round 2
Reviewer 1 Report
Thank you for addressing the previous comments. The paper is much easier to follow. I have some remaining queries/clarifications as follows:
Major comments:
1) Secretor or non-secretor status refers to the ability of an individual to secrete ABO blood group antigens in bodily fluids such as saliva. As such, it is not possible to determine ABO status in the saliva of non-secretors (will come up as A- B- and cannot assume these to be O). Analysis of the ABO status needs to be restricted the secretors only. Results and discussion need to be amended accordingly. (Apologies for not picking this up and mentioning this in the previous review of your paper.)
2) I am not convinced that the multi-variate analysis adds much to the paper, especially as ABO status can only be assessed in the secretors.
3) Table 2: Lewis phenotype row – Total Le a+b- is 45 and RV neg n=54. Please clarify this discrepancy – should be 45?
|
Le(a+b-) |
45 (13.9%) |
0 (0.00%) |
54 (19.7%) |
4) Line 300: “Similarly, none of the Le(a+b-) (Lewis negative) were infected with rotavirus.” Le(a+b-) are not Lewis negative so this should be removed. Also remove “similarly” as unclear what this is similar to.
5) Line 414-415: “Interestingly, none of the 64 children characterised as Lewis negative (Le(a+b-)) were infected by rotavirus.” Le a+b- are not considered Lewis negative – Le a-b- are Lewis negative. Not sure where the n=64 comes from.
6) Line 422: Is it possible that the high proportion of Le a+b+ may be related to the Elisa assay i.e. cross reaction with the lewis a and b antibodies?
Minor comments:
7) Table 1: Rotavirus immunization, 1 dose in RV- should be 6.14% not 61.4%.
8) Line 284: blood type B should be 9.3% (30/323) not 30/342 and blood type AB should be 7/323 not 7/333. This needs to be revised anyway in light of previous comment on ABO status in secretor negatives.
9) Line 320-322 “The mechanistic interaction between the VP4 P[8] binding to a specific site on the H1 and its precursor provides a biological association that should be explored further in future studies [24].” Move this to the discussion.
Author Response
Reviewer 1 comments, version 2
Thank you for addressing the previous comments. The paper is much easier to follow. I have some remaining queries/clarifications as follows:
Major comments:
- Secretor or non-secretor status refers to the ability of an individual to secrete ABO blood group antigens in bodily fluids such as saliva. As such, it is not possible to determine ABO status in the saliva of non-secretors (will come up as A- B- and cannot assume these to be O). Analysis of the ABO status needs to be restricted the secretors only. Results and discussion need to be amended accordingly. (Apologies for not picking this up and mentioning this in the previous review of your paper.)
Response: We beg to differ with the reviewer.
In our understanding of the A/B/H antigen system, group O may or may not have an H antigen. In the secretor / non-secretor system, the traditional definition of a ‘secretor’ is an individual who secretes blood type antigens (ie. A or B or H), in their body fluids like saliva, whereas a non-secretor does not secrete their blood type antigens into these fluids. Based on recent knowledge on the biosynthesis pathways of histo-blood group antigens (HBGAs), the current definition of ‘non-secretor’ was described as an individual who has an inactive α1,2 fucosyltransferase (FUT2) that results in the lack of any H-antigen related HBGAs, including A, B, H, Leb, LeY HBGAs (please see this publication by co-authors on this manuscript - Tan M & Jiang X. Histo-blood group antigens: a common niche for norovirus and rotavirus. Expert Rev Mol Med. 2014 Mar 10;16:e5).
Thus, the older concept of ‘non-secretor’ is no longer accurate, because non-secretor can secret Lea and/or Lex antigens (products of FUT3) in the bodily fluids. Based on the above definitions, it is clear that A and B blood types are those who have the A and B antigens, respectively, whereas O blood types are those who do not have A or B antigens. Furthermore, depending on the presence of the H-related antigens, blood type O can be further defined into a ‘secretor’ blood type O (exhibiting related antigens) or a non-secretor blood type O (with no H-related antigens). Based on this rationale, we hope that the reviewer will understand our blood type classifications and related analyses.
I am not convinced that the multi-variate analysis adds much to the paper, especially as ABO status can only be assessed in the secretors.
Response: We would like to keep the multivariate analysis. As described above, we believe the blood type classifications, especially the ABO status, are appropriate.
3) Table 2: Lewis phenotype row – Total Le a+b- is 45 and RV neg n=54. Please clarify this discrepancy – should be 45?
|
Le(a+b-) |
45 (13.9%) |
0 (0.00%) |
54 (19.7%) |
Response: We thank the reviewer for pointing out this error, which was a typographical error and has been corrected.
Line 300: “Similarly, none of the Le(a+b-) (Lewis negative) were infected with rotavirus.” Le(a+b-) are not Lewis negative so this should be removed. Also remove “similarly” as unclear what this is similar to.
Response: We thank the reviewer for this comment. The sentence has been modified as recommended and we have deleted the term “(Lewis negative)”
Line 414-415: “Interestingly, none of the 64 children characterised as Lewis negative (Le(a+b-)) were infected by rotavirus.” Le a+b- are not considered Lewis negative – Le a-b- are Lewis negative. Not sure where the n=64 comes from.
Response: The reviewer is right and the phrase “as Lewis negative” has been deleted from the sentence. “64” is a typo that has been corrected to “45”.
Line 422: Is it possible that the high proportion of Le a+b+ may be related to the Elisa assay i.e. cross reaction with the lewis a and b antibodies?
Response: The specificity of the commercial monoclonal antibodies against different HBGA types have been demonstrated in many published studies, which do not describe any cross-reactivity between any two HBGAs. The observed high proportion of Le (a+b+) may be specific Black African population related phenomenon; a previous publication revealed a similar phenomenon in Black Brazilians of African descent (see Ref [48]).
Minor comments:
- Table 1: Rotavirus immunization, 1 dose in RV- should be 6.14% not 61.4%.
Response: Thank you for detecting this discrepancy which is a typographical error with an incorrect placement of the decimal point. We have corrected the manuscript to be correct.
- Line 284: blood type B should be 9.3% (30/323) not 30/342 and blood type AB should be 7/323 not 7/333. This needs to be revised anyway in light of previous comment on ABO status in secretor negatives.
Response: Thank you for identifying this discrepancy in our numbers and calculations. We have revised the text with the correct numbers and percentages. As described above, we have retained the ABO blood type classification, so with the corrected numbers, this section remains unchanged.
- Line 320-322 “The mechanistic interaction between the VP4 P[8] binding to a specific site on the H1 and its precursor provides a biological association that should be explored further in future studies [24].” Move this to the discussion
Response: We have removed this text and added some to the Discussion section (lines 344-345).
Reviewer 3 Report
According to the reviewer's previous comments, is necessary to use a valid control to extrapolate the data obtained in the study to the population of South Africa in order to draw valid conclusions. Only symptomatic children with gastroenteritis were used in the table 2 , and therefore, the conclusions obtained are related to this population.
The authors consider that the children with diarrhea not differ significantly from healthy non-diarrheal children in that community, but the bibliography provides many examples (cited in the present manuscript) suggesting that rotaviruses and noroviruses preferentially infect secretor-positive populations.
Author Response
Reviewer 3 comments, version2
According to the reviewer's previous comments, is necessary to use a valid control to extrapolate the data obtained in the study to the population of South Africa in order to draw valid conclusions. Only symptomatic children with gastroenteritis were used in the table 2, and therefore, the conclusions obtained are related to this population.
Response: We described in our first revision the lack of “control subjects”, as this study utilized an ongoing surveillance system for diarrheal children attending either an outpatient clinic or a large hospital. The population in the catchment area of the hospital and its surrounding 23 clinics and District Hospitals are an indigenous population of Black African ethnicity, and other pilot studies have indicated the homogeneity of this population.
The authors consider that the children with diarrhea not differ significantly from healthy non-diarrheal children in that community, but the bibliography provides many examples (cited in the present manuscript) suggesting that rotaviruses and noroviruses preferentially infect secretor-positive populations
Response: We thank the reviewer for this question, and certainly there is data in the published literature that suggests rotavirus infection may be more predominant in certain populations. However, even in African studies, this data is not consistent.
Therefore, we do not believe that this issue is relevant in this small study for two reasons. First, this is a very homogenous population of Black ethnicity which feeds into the catchment area of the two study sites. The last population census was conducted in South Africa in 2011, and although a new population census was planned for 2022, these results are not available. The available data from that census describes a remarkably homogenous ethnic population. Secondly, we have been conducting diarrheal surveillance for over 25 years in this community and have never identified a population sub-group that is more or less susceptible to acute rotavirus-confirmed diarrhea